# Corporate Social Responsibility Information in Annual Reports in the EU—A Czech Case Study

**Radka MacGregor Pelikánová** 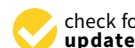

Department of Industrial Property, Metropolitan University Prague, Dubečská 900/10, 100 31 Prague 10, Czech Republic; radka.macgregor@mup.cz; Tel.: +420-725-555-312

**Abstract:** The commitment of the European Union (EU) to Corporate Social Responsibility (CSR) is projected into EU law about annual reporting by businesses. Since EU member states further develop this framework by their own domestic laws, annual reporting with CSR information is not unified and only partially mandatory in the EU. Do all European businesses report CSR information and what public declaration to society do they provide with it? The two main purposes of this paper are to identify the parameters of this annual reporting duty and to study the CSR information provided by the 10 largest Czech companies in their annual statements for 2013–2017. Based on legislative research and a teleological interpretation, the current EU legislative framework with Czech particularities is presented and, via a case study exploring 50 annual reports, the data about the type, extent and depth of CSR is dynamically and comparatively assessed. It appears that, at the minimum, large Czech businesses satisfy their legal duty and e-report on CSR to a similar extent, but in a dramatically different quality. Employee matters and adherence to international standards are used as a public declaration to society more than the data on environmental protection, while social matters and research and development (R&D) are played down.

**Keywords:** corporate social responsibility; environment; employment; R&D; annual reports; financial and non-financial statements; competition

---

## 1. Introduction

Despite the sui generis status of the European Union (EU) and the alleged chronic EU blurring of the distinction between truth and reality [1] and between law and politics [2] in a business and competition context [3], the EU is committed to Corporate Social Responsibility (CSR). Indeed, this commitment is included in both its 10-year strategy Europe 2020 as well as in EU law, which sets a legislative framework for e-reporting of both financial and non financial statements. Since EU member states freely, or less freely [4], reflect and further develop this framework by their domestic laws, and the resulting systems aim at, but do not command, free, centralized, electronic, periodic and detailed financial and non-financial reporting. Although undoubtedly the importance of business ethics and CSR keeps growing [5] (p. 254), CSR reporting is only partially mandatory across the EU and the types, quality and quantity of data to be provided about CSR is not regulated in a unified manner. Therefore, a rather complex legislative setting emerges with a real application with relevance for the majority of, if not all, businesses in the EU. However, its understanding is essential in order to address the status quo of CSR reporting in the EU, and in particular in the Czech Republic. This leads to the need this paper to have two main and highly important purposes—to determine this EU and Czech legal framework and to assess its real-life application. Namely, an understanding of the current exact parameters of the CSR reporting in a selected jurisdiction within the EU, such as the Czech one, needs to be established first, so as to establish a foundation for the performance of a national case study regarding the reality of the CSR reporting in such a jurisdiction.

Firstly, since there are just a few holistic legal studies exploring the exact parameters of the current legislative framework regarding the CSR reporting in the EU and EU member states, such as the Czech Republic, via annual reports, it is vital to do so. Despite the EU's commitment to a single internal market, to the transparency and accessibility of business documents, such as annual reports, and to CSR and the general perception of their importance, academia, businesses and the public at large remain in the dark regarding the exact extent of the legal duty in the EU and the Czech Republic. In other words, do Czech businesses have to e-file regularly their annual statements with CSR information with their business registers and, if so, what does this CSR information have to include and entail? One academic stream proposes that the competition reflected by the CSR e-reports might be perceived as information in the public sphere, i.e., a public good, which reflects the positive social orientation of people [6,7] and develops the much needed critical awareness among the public regarding positive and negative impacts of business conduct [8–10]. Arguably, it leads neither to a rivalry nor to excludability [11] and supports competitiveness [12,13]. Indeed, e-reporting about CSR can assist in the prevention of economic crises [14,15] and be a positive factor for proper competition and even coopetition [16,17], especially in the field of intellectual property and research and development (R&D) [2,18,19].

Naturally, this perception is fully in compliance with the vision of the internal pro-integration EU tandem—the European Commission and the Court of Justice of the EU [19–21]. However, other studies are less conclusive about the reconciliation of the EU values with CSR [22], the determination of competitors to adopt altruistically CSR [23], the reports about it, and the general endorsement by the community promoting social norms [24]. Some authors even propose that empirical analysis shows such a vast complexity of CSR reporting, and its conversion, that they might both be spontaneous as well as regulated, and the realization, thereby, is at the edge of feasibility [25], if not even beyond [26]. Hence, there is an ambiguity about the (lack) of both the legal duty and social duty with respect to CSR and its e-reporting as such, and perhaps even about its legitimacy [27]. This leads to controversies and misunderstandings about whether CSR information has, should, does not need to, or even should not be included in annual reports filed electronically and freely available.

Secondly, even more complex controversies emerge after the determination of the existence of this duty, namely, what is to be included and ultimately what is included in the reality. Boldly, once the existence of this duty is clarified, an even more important issue emerges, namely, what CSR information is, in real life, provided in the annual statements and what public declaration to society is achieved by it? There are just a few, generally national and strongly fragmented studies, and these with inconclusive propositions about the reality of CSR reporting [20,28–32]. Generally, they propose that the understanding of the extent, format and form of CSR varies considerably [27,29], that the impact of the CSR on performance varies [33], and even tensions and conflicts between various types and categories of CSR information exist [26]. Some studies even suggest that the satisfaction of the legal duty of the CSR reporting is rather fictive because the majority of, e.g., Czech, companies do not provide quantitatively and qualitatively appropriate CSR information in any of the CSR types, i.e., neither about the environmental protection nor employment matters, etc. [28]. Nevertheless, new studies establish that the perception of the CSR, which is obviously intimately linked to the involved business ethics, has significantly developed and the commitment to CSR and its reporting has been reinforced and even partially standardized via ISO norms, in particular within the newer EU member states [9,10,34].

Due to this lack of many prior case studies [20,28–32], and because those that exist bring, based on linear regression, very interesting suggestions about the CSR and its dimensions, such as employees and environmental matters [31], the realization of a Czech pioneering national case study about the categories, aka types, quantity and quality of CSR information provided in annual statements centrally filed, and accessible online, is highly desirable. Since it is proposed that both the company size and financial performance affects the CSR and CSR reporting much more than its belonging in a particular type of industry [31], the selected sample includes the 10 largest Czech companies by revenue and their annual reports, available online via the central e-portal, for the period 2013–2017. This is dynamically

and comparatively explored and scrutinized with respect to the types of CSR information—research and development (R&D) leading to innovations [13,35], environmental protection, employment matters, and others. The high compliance rate of 90% regarding CSR information in the e-published annual reports and the often problematic quality and reduced quantity of the CSR information provided might suggest that the CSR information is perceived as a legal duty to be formally observed only.

Based on legislative research and the teleological interpretation of the current EU legislative framework with Czech particularities, plus the Czech case study, the original meta-analysis implies that Czech businesses, at least large companies, satisfy their legal duty and e-report on CSR to a similar extent, but in a dramatically different quality. The preference for both the form over the content and for certain types of CSR information is accompanied by other indices about how companies use CSR information in their e-published annual reports as a public declaration. Employee matters and adherence to international standards are used as a public declaration to society more than data on environmental protection, while social matters and R&D are downplayed. Indeed, there is even an occasional lack of perception of R&D as an element or factor of CSR. These are critical points deserving further study.

## 2. Conceptual and Legislative Framework

Today's global, complex, highly competitive and heavily digitalized society has brought forth many challenges for European integration [1–4,36]. The EU and its member states attempt to address them through various instruments designed to promote competitiveness, transparency and communication, with different levels of effectiveness and efficiency [19,21,27]. The prior proposition that the implementation of the concept of CSR is not obligatory [27] and that it will be freely embraced by subjects progressively recognizing its benefits, has been modified by clear indices and even demands of the EU law, and often even national laws, making the CSR concept and basic (not detailed) reporting about its realization compulsory for certain subjects.

The understanding and appreciation of the publication of CSR information in annual reports in the EU, and in particular in the Czech Republic, logically has two prerequisites. First, the conceptual framework, including contextual priorities and underlying concerns, needs to be recognized. Second, how this conceptual framework is vested and projected in the applicable legislative framework, i.e., both by the EU law and Czech laws, has to be assessed. Boldly, it is mandatory to recognize whether businesses have to file and make centrally and electronically accessible their annual reports and whether these annual reports have to include CSR information.

### 2.1. Conceptual Framework

The interaction of law and business, and, more specifically, the interaction of legal, moral and social obligations with regard to business conduct, is full of contradictions [36]. Since the 1970s, these contradictions have led to a realization that there is a conflict between the commitment to the promotion of globalized economic growth and the issue of increasing world population needs, along with the degrading ecological situation [37]. The pendulum of balance has importantly moved and led to the burning question posed, among others, to and by the United Nations (UN), namely how to achieve global prosperity without environmental deterioration in the world [32], in both developed and developing countries. This led to the origins of CSR, embedded in the proclamation 'Our Common Future—A global Agenda for Change', prepared by the Brundtland Commission which was published in 1987 as the United Nations (UN) Annex to document A/42/427 and which was followed by the UN Agenda 21 and UN Resolution A/RES/60/1 from 2005 (Resolution 2005) further developing the idea of sustainability at the international level [27]. The initial focus on sustainability has been progressively paralleled by corporate responsibility concerns. Namely, the sustainability concept as a systematic and visionary tool governed predominantly by soft law has co-existed with the corporate responsibility concept as rather a normative and moral tool regulated by hard law, until they merged into CSR [38]. In the digital setting of the initial decades of the 21st century, this is underscored by

Art.36 of the Resolution 2005, which demands equity and transparency of financial and business systems, and by Art.49 of Resolution 2005, which envisages three mutually reinforced pillars for sustainable development—economic development, environmental protection and social development. Moving down to the ultimate addressees, the CSR became for businesses the synonym for the transition from the "profit only" emphasis [10] to "profit, people, planet", leading to a focus on the combination of economic prosperity linked to innovations, environmental quality and social improvements [12,22]. Baldly stated, CSR makes businesses responsible, perhaps even liable, not only to their shareholders, but as well to other categories of their stakeholders [39].

The combined effect and impact of the economic crises, the failed Lisbon Strategy 2000–2010 and other negative events have significantly contributed to the shifting of the focus to the development of regulations on financial, corporate management, corporate governance and liabilities matters, even in the EU [3,14]. Naturally, this trend includes also the discussion, proposition and even legal regulation of minimum standards of CSR and reporting about it [25] with respect to the aforementioned three pillars of Resolution 2005—economics linked to R&D, environmental linked to environment protection, and social linked to employment and other matters. The CSR and related business ethics dimension are progressively perceived as general directions to create a future world which will fairly and sustainably balance economic, environmental and social factors [40] in the context of the highly competitive knowledge economy [13].

Certain studies from the EU propose a growing interest in CSR and suggest that this leads to some pressure for companies to recognize, apply and report regarding CSR [20,31] and contribute to the enhancement of awareness [8], while other studies are less conclusive [25] and even suggest a reluctance to do so [28]. This is further magnified by the ongoing issue of balancing between the neoclassical equalization of the levels of development between jurisdictions of the EU and the process of EU member states' internal divergences [11]. So it remains unclear and open, both on the legislative and academic levels, regarding what, exactly, quantitatively and qualitatively should be included in the annual reports of the qualified companies about the CSR. Similarly, it is open to assess the exact dimension of the strategic role of disclosure by businesses, regardless of whether they have the legal form of a company or another legal form [6,8,19]. Namely, for some businesses the disclosure of relevant information, including CSR data, is an opportunity taking on the shape of a popular marketing tool, while, conversely, for other businesses it is a threat of an unknown or even of a dangerous source for self-incrimination or unfair practices [19,26].

## 2.2. Legislative Framework

The current EU, its law and strategy, aka Europe 2020, focuses on the single internal market in which smart, sustainable and inclusive growth takes place [41] and where the technological potential [13,42,43] and fair competition [44] should develop and could lead to the EU world trade leadership [45], as repeatedly proposed by the European Commission [46] and skeptically observed by others [47]. This is implied by the EU primary law, especially the Treaty on the EU (TEU) and the Treaty on the Functioning of the EU (TFEU), as well as by EU secondary law, represented predominantly by regulations and directives. The legislative framework directly covering the annual e-reporting about CSR rests on three pillars—two directives on statements and reports to be prepared and filed by businesses and one regulation about the manner of e-publication. This EU law penetrates into national laws of the EU member states [4], and these states add to it their own certain particularities.

The first pillar is the Directive 2013/34/EU on annual financial statements (Directive 2013) which specifically sets a CSR reporting duty for large businesses. Pursuant to its Art.19a(1), public interest entities with over 500 employees shall include in the management report a non-financial statement containing information on environmental, social and employee matters, respect for human rights, anti-corruption and bribery matters, etc. This duty set by the EU law may be exempted based on Art.19a(4) which allows EU member states to exempt undertakings which e.g., provide this information on their websites. However, if public interest entities with over 500 employees are not exempted

by national law then, based on Art.30(1), they have to publish annually their balance sheet, annual financial statements and the management report, i.e., basically they have to file their annual reports including the CSR information. However, no further regulation on the EU level is provided and, pursuant to comparative studies, the CSR concept is not fully obligatory in EU member states [48] and the reporting practice with respect to its content is quite diversified across the EU as well as across industries [25].

Hence, EU law sets a legal principle that public interest entities with more than 500 employees must include CSR in their management report, and national laws can either expand this duty, by expanding the information or the publication methods, or reduce this duty, by providing exemptions and exceptions. Namely, each EU member state nationally adjusts this legal principle, i.e., it adds to the general lines of the EU legal framework based on the Directive 2013 some national particularities features, which can often lead to more fragmentation and ambiguity and seldom leads to more harmonization and clarity. Regarding the Czech national law provisions with respect to CSR and its reporting, the fundamental statute is Act No. 563/1991 Coll., on accounting (Act 1991), which, among other items, regulates by its Art.18 et foll. the compulsory content of financial and other statements. In addition, Act 1991 requires, via Art.21, that larger businesses and companies have their statements, including annual reports, verified by a public auditor. Although Act 1991 regulates by Art.21 the compulsory content of the annual report and demands both financial statements as well as CSR information, namely about R&D, environmental protection and employment relationships, it does not further clarify the extent and depth of such CSR information, i.e., it does not explain the expected quantitative and qualitative dimension of the CSR information. Hence, Czech law complies with the EU law without further clarifying it, and there is no doubt that Czech public interest entities with over 500 employees must include "some" CSR information in their annual reports, i.e., to provide "some" data on non-financial key performance indicators, including information relating to the environment and employee matters [49].

The second pillar is Directive 2017/1132/EU relating to certain aspects of company law (Directive 2017), which repeals, among others, Directive 2009/101/EC September 2009 on the coordination of safeguards which, for the protection of the interests of members and third parties, are required by member states. Directive 2017, by its Art.13 et foll., provides a general framework for the disclosure and publication of documents in commercial registers. Instead of terms such as undertaking or public interest entities, as used by Directive 2013, Directive 2017 provides a list of its addresses by its Annex II. For the Czech Republic, Annex II of Directive 2017 determines as the addresses of the publication duty of Art.13 et foll. of Directive 2017 the limited liability company (s.r.o.) aka private limited company, and the shareholder company (a.s.) aka public limited company. Documents to be disclosed by these addresses are listed by Art.17 and they include accounting documents for each financial year to be published pursuant to Directive 2013. Regarding the form, Art.16 specifies that each EU member state has to keep a central company register and that all documents required by Art.14 are to be kept on the file. Hence, despite the terminology that is not perfectly matching and pre-requirement setting, it can be summarized that the EU law via Directive 2013 and Directive 2017 generally demands that larger public entities (greater than 500 employees plus s.r.o. or a.s. form) have to include the CSR information in their reports, and that these reports have to submitted to the central company register in order to be kept in each subject file and potentially available for third parties.

Czech national law is fully compatible with this (see above). Indeed, Act 1991 regulates via Art.20–21a the compulsory publication of corporate documents and various statements for entities registered in public registries. Another Czech statute, Act No. 304/2013 Coll., on public registries of legal entities and natural persons (Act 2013), provides, by Art.1–3, that the public registries are maintained electronically by courts and by Art.42 that all companies and corporations are to be registered in the Commercial Register. Further, Act 2013 specifies by Art.66 documents to be filed with the Collection of documents, i.e., provides a list of documents to be filed with the court keeping the

Commercial Register and placed in the Collection of documents. From the perspective of the CSR reporting, it is critical that these documents are, by the operation of Art.3, made freely available to the public in an electronic form and pursuant to Art.66, they include annual reports and final statements provided that this duty is envisaged by the Act 1991. In sum, each and every Czech company or corporation needs to be registered in and have filed corporate documents and various statements with Commercial Register. For companies with at least 50 employees or annual turnover of over CZK 80 million etc., this duty includes as well the filing of annual reports with final statements approved by the auditor. Pursuant to Art.42, Commercial Register with Collection of documents are maintained by the Commercial court and are freely electronically accessible. Unlike with other EU member states, the free e-publication of annual reports is an actuality in the Czech Republic and documents filed with the Commercial Register and placed in the so called Collection of documents are posted in a pdf format on the platform placed on the Czech country code domain "TLD.cz" (justice.cz) [50]. Table 1 summarizes the key provisions of this legal framework.

**Table 1.** Key provisions constituting the legal framework for Corporate Social Responsibility (CSR) e-reporting in the Czech Republic.

| Source | Content |
| --- | --- |
| Directive 2013 | *Art.19a(1) Non-financial statement* <br> *"1. Large undertakings which are public-interest entities exceeding on their balance sheet dates the criterion of the average number of 500 employees during the financial year shall include in the management report a non-financial statement containing information to the extent necessary for an understanding of the undertaking's development, performance, position and impact of its activity, relating to, as a minimum, environmental, social and employee matters, respect for human rights, anti-corruption and bribery matters . . . "* |
| Directive 2017 | *Art.13 Scope* <br> *"The coordination measures prescribed by this Section shall apply to the laws, regulations and administrative provisions of the Member States relating to the types of company listed in Annex II."* <br> *Art.14 Documents and particulars to be disclosed by companies* <br> *"Member States shall take the measures required to ensure compulsory disclosure by companies of at least the following documents and particulars . . . the accounting documents for each financial year which are required to be published in accordance with Council Directives . . . "* <br> *Art.18 Availability of electronic copies of documents and particulars* <br> *"Electronic copies of the documents and particulars referred to in Article 14 shall also be made publicly available through the system of interconnection of registers"* <br> *Annex II Types of Companies referred to in Articles 7(1), 13, 29 . . .* <br> *—Czech Republic: společnost s ručením omezeným, akciová společnos* (private limited company—limited liability company (s.r.o./Ltd.) and public limited company—shareholder company (a.s./SA). |
| Act 1991 | Art.20 duty to have final statements verified by an auditor extends to companies with at least 50 employees or a turnover over CZK 80 million or . . . <br> Art.21 companies with the duty to have financial statements verified have as well the duty to prepare annual reports . . . Annual reports have to include both financial and non financial information including about R&D, environment and employment matters . . . |
| Act 2013 | Art.2–3 Commercial Register along with Collection of documents are kept by courts and are freely available in a digital format <br> Art.42 all companies and corporations to be registered with Commercial Register <br> Art.66 document to be filed in the Collection of documents include annual reports and final statements as stated in Act 1991 |

Source: Prepared by the author based on eur-lex.

Hence, the parameters of the annual CSR duty for Czech larger public entities, being both more than 500 employees and s.r.o. (private limited company aka Ltd.) or a.s. (public limited company aka shareholder company), are pretty clear—they have to include the CSR information in their reports, and these are filed in the Collection of documents and ultimately available in pdf to the public at large for free. Yet this clear form and format setting is not matched by clarity regarding the content,

i.e., basically both EU and Czech law do not specify in detail the minimum threshold of the CSR information to be provided, as a result of which even a very brief and superficial note about the CSR might be considered as satisfactory. In sum, the quantity and quality of the CSR information provided is not determined, since neither the EU law nor Czech law regulates clearly and expressly regarding this manner.

Pillar number three is Regulation (EU) 2015/884 establishing technical specifications and procedures required for the system of interconnection of registers, which created the electronic system of interconnection of registers called the Business Registers Interconnection System (BRIS). Within the BRIS, data critical for financial accounting, tax and even managerial accounting [51,52] is migrated from national Business Registers to the European Central Platform and available at the e-Justice Portal placed on a sub-domain of the EU top-level domain "TLD.eu" (e-justice.europa.eu) [53]. Hence the search at the e-Justice Portalallows for a central search based on a name or a registration number within all migrated data or a search within a national Business Register, and it can both establish and/or eliminate a competitive advantage in our information digital era. However, annual reports do not belong to the compulsory content data and so can, but do not need to, be accessible for free or the access fees are not exceeding the administrative costs. In this respect, Czech national law goes beyond it and makes annual reports, including CSR information, freely electronically available. However, the data is not perfectly migrated via BRIS and so a search for Czech annual reports has rather to be done while directly using the Czech national platform, i.e., rather via justice.cz than e-justice.europa.eu.

## 3. Materials and Methods

Both of the two main raisons d'être of this paper focus on (i) the parameters of the annual reporting legal duty with respect to CSR and (ii) the realization of this by Czech companies. This entails many tools and processes going from a critical and partially descriptive analysis of the legislative acts and secondary academic sources from various jurisdictions to a field search and case study. The cross-disciplinary and multi-jurisdictional nature of the exploration requires holistic processing with the employment of meta-analysis [54]. The interplay of economic, legal and technical aspects shapes the focus, targeting both qualitative and quantitative data and entailing deductive and inductive aspects of legal thinking [55]. Thus, the quantitative research and data is complemented by qualitative research, along with critical closing and commenting, and refreshed by Socratic questioning [56].

Regarding the first line of the dual purpose, this implies that both legislative and academic literature research, focusing on the sources from the EU, Czech and other EU member states' jurisdictions, needs to be done and the mined data properly teleologically interpreted, while fully recognizing the importance of the purposive and mischief approach [1,27] The current EU legislative framework regarding annual reporting, in particular e-reporting including CSR information, is revealed along with Czech particularities. Hence the first line relies predominantly on the legislative research and teleological interpretation of EU legislative primary and secondary sources, represented by the three pillars' legal construction, consisting of two directives and one regulation, and is further clarified by recent secondary academic sources ranked and classified in the Web of Science (WoS) and Scopus databases.

Regarding the second line of this paper's two goals, a pioneering Czech case study was performed while using a representative sample of the ten largest Czech companies and their annual statements for 2013–2017. The case study dealt with the Czech Republic because, as indicated above, the legal parameters for CSR reporting vary across all EU jurisdictions and only the Czech particularities are covered by this paper. The representative sample of Czech businesses was selected in the manner described so as to obtain homogenous subjects offering real and verifiable data. Boldly, to use well known businesses which are subject to the legal CSR reporting duty and about which the information can be checked via justice.cz and even double checked (their own www). Hence the criterion used was 'largest' as the total of annual revenues in the last observed years, i.e., 2016 and 2017. Interestingly enough, all of these top companies have annual revenues exceeding CZK 50 billion, but their assets

ranged from CZK 10 billion to CZK 600 billion and their net income from "red numbers" to a very black number of CZK 20 billion. The list below indicates their name, identification number and their field of industry.

1. Škoda Auto a.s., ID 00177041—automobiles;
2. ČEZ, a.s., ID 45274649—electricity;
3. Agrofert, a.s.—conglomerate, agriculture;
4. RWE Supply & Trading CZ a.s., ID 26460815—oil and gas;
5. Foxconn Technology CZ, s.r.o., ID 27516032—consumer electronics;
6. UNIPETROL, a.s., ID 61672190—chemicals;
7. Hyundai Motor Manufacturing Czech s.r.o., ID 27773035—automobiles;
8. ČEPRO, a.s., ID 60193531—oil and gas;
9. Continental Automotive Czech Republic s.r.o., ID 62024922—automobiles;
10. Finitrading a.s., ID 61974692—iron, steel, finances.

All of these companies have a legal form of either a shareholder company or a limited liability company, they employ more than 500 employees, and file their annual reports with the Czech Commercial Register, which makes them freely available in the electronic format pdf via the national portal available via justice.cz and the EU system BRIS. The case study did not include smaller companies and so was not impaired by incomplete, missed or not verifiable data, i.e., all companies used for the Czech case study are known, have www pages presenting information about them and including even their reports, and they all are subject to the prescribed legal duty, and, at least prima facie, satisfy it.

Hence, the sample was selected regardless of the field of industry and all the included companies meet the conventional requirements for e-reporting. Since the research via BRIS and the national Czech Commercial Register yields annual reports regarding all of these companies for the entire period of 2013–2017, the critical and comparative exploration of 50 annual reports and their data about the type, extent and depth of CSR information could be performed. The data on CSR was classified by their type and these types were set based on Directive 2013, i.e., R&D, environmental protection, environmental matters and others. These four types are further described in the recital 26, Art.19a, Art.29a of the Directive 2013, in explanations provided by the European Commission [46] and in ISO 26000. Considering the focus on the Czech Republic, it is pivotal to underline that the Czech national law deals exactly with these four dimensions, see Act 1991 and especially its Art.21. Unfortunately, this Czech national provision does not provide any further descriptions and merely satisfies itself by demanding annual reports with non financial statements about "activities in the field of R&D, activities in the field of environment protection and in employment relationships". Hence these four dimensions are officially and explicitly recognized by Czech law and CSR information in annual reports has to entail them, but there is no clear rule about the expected exact extent and depth of this information and so there is used the reliance on the indication provided by the EU law (Directive 2013) with the above mentioned explanations and ISO norms.

The quantitative aspect was addressed by calculating the total number of pages, i.e., how many pages long was the entire annual report, on how many pages was the CSR information contained and on how many pages each type of the CSR information (R&D, environmental protection, environmental matters and others) was included. The quantitative criterion of pages rather than sentences was selected due to the linguistic, especially stylistic and pragmatic semiotic, particularities of the Czech language belonging to the Slavic language group. The qualitative aspect was addressed by the holistically manual approach employing a simplified Delphi method. Namely, each and every one of these 50 annual reports was carefully read through by three experts on corporate matters including reporting (EDC, LM and RKM, i.e., none of these three experts was the author of this article) while following a universal set of guidelines and simple questionnaires prepared by the author. All three experts master both Czech and English, have college degrees, experience with annual reporting, at least

20 years of executive job experience and a strong law and/or economic background. Two of them are women and one is a man. Hence, their replies met the expertise expectations. These first-round replies were processed by the author and based on them the author prepared a summary which was communicated to these three experts for the second round. Thereafter, they made few changes with respect to their prior answers and sent their updated replies to the author. This data, generated from the second round, was used for the paper.

Specifically, based on these guidelines and questionnaires, each of these three experts categorized the provided CSR information (+) or (++) or (+++). The guidelines required ranking as no more than general information (+) all universal and proclamation-type statements lacking a relationship to real and controllable actions or omissions; to ranking as more developed and concrete information (++) all statements leading to a single real and controllable action or omission or participating on general CSR trends; and as robust information (+++) all statements about real and controllable actions culminating in an exemplary CSR behavior linked to the particular business and that was made public and regarding the existence of which is beyond any doubt. Hence, plain statements such as "we recognize the importance of environmental protection" were (+), more developed statements such as "we make our products in an environmental friendly manner by using this and not that" were (++) and information about pro-active tangible CSR behavior, such as "although we provide services and do not directly pollute the environment, we decided to revitalize the park by the daycare XY by planting 100 trees and by being responsible for the ongoing care for them . . . and because of this and other acts, we received an award . . . (or see photos of this park below)" were (+++). Hence, each of these 50 annual reports has been seen and ranked by three experts independently. The results from the first round was processed by the author and resent to the experts who then provided adjusted results in the second round. These results were compared and, in the case of still different results (one expert giving more or less ++ than others), this then led to these three experts conferring with the author and together agreeing about the proper ranking. In addition, the analyses performed by these three experts, while studying these reports and later confronting their ranking, allowed for extracting critical statements, quotes and declarations to be demonstratively indicated in the tables below, i.e., Tables 2–11.

## 4. Results

The e-reporting on CSR is the hallmark of a current relationship and interaction of a wide spectrum of stakeholders in the internal single market [32,57], which is only partially covered by mandatory, expressed and explicit legal norms [27,38]. Ultimately, the above described legislative framework includes ambiguities, terminological imprecision and even vacuums and the alleged split between old and new EU member states reappears [58], in particular in the light of Europe 2020 [59–61]. Nevertheless, one can argue that it implies a general rule that shareholder companies and limited liability companies with more than 500 employees have to include CSR information in their annual reports and file them with their national Commercial Registers.

Czech law extends this general legal duty generated by Directive 2013 and Directive 2017 and demands electronic filing. This leads to the free availability of annual reports and ultimately becomes a part of the data available via BRIS. The selected 10 largest Czech companies satisfy the given criteria and are subject to this legal duty, i.e., their annual statements with CSR information are to be available via BRIS.

It is extremely interesting to make a pioneering case study involving annual reports of the 10 largest Czech companies for 2013–2017 and conduct research about whether they provided CSR information and, if yes, of what kind (R&D, environmental protection, employment matters, others) and in what quantity and quality (on the scale from + to +++ as indicated in 3. Materials and Methods), and whether the very wording or its general spirit provides hints and indices able to be considered as declarations. The set of tables below, i.e., Tables 2–11, addresses these questions and provides an insight that is truly original.

**Table 2.** Annual reports of Škoda Auto a.s., ID 001 77 041.

| Year | CSR/All | R&D | Environm. | Employm. | Others |
|------|---------|-----|-----------|----------|--------|
| | Pages/Pages | Pages/Quality | Pages/Quality | Pages/Quality | Pages/Quality |
| 2013 | 7/192 | 2/+ | 2/++ | 2/++ | 1/+ |
| 2014 | 8/126 | 2/+ | 2/++ | 2/++ | 2/+ |
| 2015 | 6/128 | 1/+ | 1/++ | 2/++ | 2/+ |
| 2016 | 6/112 | 2/+ | 1/+ | 2/++ | 1/+ |
| 2017 | 9/148 | 1/+ | 2/++ | 3/++ | 3/+ |
| Quotes—Declaration | " . . . won an award at the "TOP Responsible Company" competition . . . surveys of public opinion once again rated ŠKODA AUTO one of the most popular companies in the Czech Republic . . . it also finished first in the CZECH TOP 100 and Czech 100 Best rankings and is therefore . . . " | | | | |
| Comments | Many declarations are highly subjective and e.g., the laudatory results regarding employments are caused by the fact that Škoda Auto has a lot of employees sending their votes in to the 100 Best competition and making their employer win regardless to objective achievements (100 Best does not have any criteria, i.e., employees are just voting for a selected employer) | | | | |

As indicated in Table 2, Škoda Auto includes a management report with the CSR information in its annual report and has a special CSR section, which is rather short (3–7% of the total number of pages) and only partially goes into depth. The most discussed topics are employment matters. Interestingly, these annual reports rely heavily on external and extrinsic evidence and refer to achievements having a CSR impact, such as winning awards. The reading and orientation in these annual reports is easy and intuitive.

**Table 3.** Annual reports of ČEZ, a.s., ID 452 74649.

| Year | CSR/All | R&D | Environm. | Employm. | Others |
|------|---------|-----|-----------|----------|--------|
| | Pages/Pages | Pages/Quality | Pages/Quality | Pages/Quality | Pages/Quality |
| 2013 | 13/300 | 4/+++ | 3/++ | 3/++ | 3/++ |
| 2014 | 12/326 | 3/+++ | 3/++ | 3/++ | 3/++ |
| 2015 | 15/329 | 4/+++ | 5/+++ | 4/+++ | 2/++ |
| 2016 | 13/332 | 3/+++ | 5/+++ | 3/+++ | 2/++ |
| 2017 | 14/356 | 5/+++ | 4/+++ | 3/+++ | 2/++ |
| Quotes—Declaration | " . . . Emission limits observed . . . EU Emission tickets commercialized . . . Reduction of working hours per week to 37.5 h . . . Education programs . . . International and national R&D projects . . . "— ČEZ presents itself as very active in all aspects of CSR | | | | |
| Comments | ČEZ is a "state" company and the extent of CSR data is influenced by the field of industry as well as the "state" feature and related need to address the (lack of) political impact. | | | | |

**Table 4.** Annual reports of Agrofert, a.s., ID 261 85 610.

| Year | CSR/All | R&D | Environm. | Employm. | Others |
|------|---------|-----|-----------|----------|--------|
| | Pages/Pages | Pages/Quality | Pages/Quality | Pages/Quality | Pages/Quality |
| 2013 | 1/109 | >1/+ | >1/+ | >1/+ | 0 |
| 2014 | 1/110 | >1/+ | >1/+ | >1/+ | 0 |
| 2015 | 3/117 | 1/+ | 1/+ | 1/+ | 0 |
| 2016 | 3/92 | 1/+ | 1/++ | 1/++ | 0 |
| 2017 | 3/114 | 1/+ | 1/++ | 1/++ | 0 |
| Quotes—Declaration | " . . . a lot of certificates about environment protection . . . strictly complies with the labor law"—Agrofert does not convey a message about its committed to CSR | | | | |
| Comments | Agrofert presents certain information in a misleading or confusing manner. | | | | |

As indicated in Table 3, ČEZ includes several sections on CSR information in various sections in its annual report. The combined CSR information is rather short (3–4%), but goes into sufficient depth. The most discussed are R&D matters. Interestingly, these annual reports rely heavily on discussions of

diverse complex strategies and thus reading them and comprehending the meaning in them is not layman friendly.

As indicated in Table 4, Agrofert includes just a few notes about CSR and they take in total very little space (1–3%) and do not go into any depth. No CSR matters are really discussed and the spirit of the annual reports rather undermines the CSR's importance. The observation of the labor law is presented as if it was done as a CSR favor, and obviously such a statement is misleading, if not directly wrong.

**Table 5.** Annual reports of RWE Supply & Trading CZ a.s., ID 26460815.

| Year | CSR | R&D | Environm. | Employm. | Others |
|---|---|---|---|---|---|
| 2013 | 1/41 | >1/+ | >1/+ | >1/+ | 0 |
| 2014 | 2/60 | >1/+ | >1/+ | 1/+ | 0 |
| 2015 | 4/55 | >1/+ | >1/+ | 3/++ | 0 |
| 2016 | 4/57 | >1/+ | >1/+ | 3/++ | 0 |
| 2017 | 3/62 | >1/+ | >1/+ | 2/++ | 0 |
| Quotes—Declaration | " . . . Diversity Talks . . . No R&D . . . Involvement with Dow Jones Sustainability Index and Carbon Disclosure Project . . . "—REWE presents itself as an open minded and no discriminating employer | | | | |
| Comments | Surprisingly and in contrast to its orientation, RWE skips the R&D field. | | | | |

As indicated in Table 5, RWE includes just a few paragraphs about CSR and they take in total a small space (3–7%), but still sufficiently detailed information is provided. The rather short CSR information is caused by the manner of the structure of the holding company, i.e., more CSR information is provided in annual reports of its daughter companies. The top CSR matters discussed in the annual reports of RWE deal with employment.

**Table 6.** Annual reports of Foxconn Technology CZ s.r.o., ID 27516032.

| Year | CSR | R&D | Environm. | Employm. | Others |
|---|---|---|---|---|---|
| 2013 | 2/40 | >1/+ | >1/++ | 1/+ | >1/+ |
| 2014 | 2/41 | >1/+ | >1/++ | 1/+ | >1/+ |
| 2015 | 2/41 | >1/+ | >1/++ | 1/+ | >1/+ |
| 2016 | 2/43 | >1/+ | >1/++ | 1/+ | >1/+ |
| 2017 | Not available | | | | |
| Quotes—Declaration | " . . . Environment protection and the observance of ISO 14001 . . . Anti-discrimination employment politics . . . Insurance for employees . . . Ethical codex . . . PR actions—activities for children."—Foxconn presents itself as active in all aspects of the CSR. | | | | |
| Comments | Foxconn underlines the relation between its CSR and ethical concerns. | | | | |

**Table 7.** Annual reports of UNIPETROL, a.s., ID 61672190.

| Year | CSR | R&D | Environm. | Employm. | Others |
|---|---|---|---|---|---|
| 2013 | 6/223 | >1/+ | 2/+ | 2/++ | 1/+ |
| 2014 | 6/247 | >1/+ | 2/+ | 2/++ | 1/+ |
| 2015 | 6/214 | >1/+ | 2/++ | 2/++ | 1/+ |
| 2016 | 7/207 | 2/+ | 2/++ | 2/++ | 1/+ |
| 2017 | 7/198 | 2/++ | 2/++ | 2/++ | 1/+ |
| Quotes—Declaration | " . . . The members of the Unipetrol Group are aware of their responsibility to all their stakeholders—their employees, customers, shareholders, business and social partners, and society. By means of this Code of Ethics they undertake to comply with clear principles forming a basic framework for the business and social conduct, and for the creation of the corporate culture . . . "—Unipetrol presents itself as a business engaging in the CSR in a wide manner | | | | |
| Comments | Unipetrol underlines the relation between its CSR and ethical concerns . . . | | | | |

As indicated in Table 6, Foxconn includes just a few paragraphs about CSR and they take in total little space (3–5%), the only type of the CSR discussed in some detail covers employment matters. The adherence to standardized norms and projects is underlined.

As indicated in Table 7, Unipetrol has a special short chapter on CSR in its annual reports (1%) which focuses on education, volunteering, donations and environment protection. Environment and employment matters are developed in the following parts of the annual reports, but not directly in the CSR chapter. The Code of Ethics is emphasized.

**Table 8.** Annual reports of Hyundai Motor Manufacturing Czech s.r.o., ID 27773035.

| Year | CSR | R&D | Environm. | Employm. | Others |
|------|-----|-----|-----------|----------|--------|
| 2013 | 3/48 | >1/+ | 1/+ | 1/+ | 1/+ |
| 2014 | 3/58 | >1/+ | 1/+ | 1/+ | 1/+ |
| 2015 | 3/62 | >1/+ | 1/+ | 1/+ | 1/+ |
| 2016 | 3/64 | >1/+ | 1/+ | 1/+ | 1/+ |
| 2017 | 1/31 | >1/+ | >1/+ | >1/+ | >1/+ |
| Quotes—Declaration | " . . . Eco Management and Audit Scheme . . . Management quality ISO 9001 . . . Employment Safety OHSAS 18001 . . . .Project "Good Neighbor" . . . National Quality Price . . . Endowment Fund Hyundai . . . "—Hyundai presents itself as an multicultural and open-minded international business. | | | | |
| Comments | Hyundai CSR statements do not seem convincing and its labor disputes undermine its (alleged) commitment to employment concerns. | | | | |

**Table 9.** Annual reports of ČEPRO, a.s., ID 60193531.

| Year | CSR | R&D | Environm. | Employm. | Others |
|------|-----|-----|-----------|----------|--------|
| 2013 | 7/221 | 1/+ | 2/+ | 2/++ | 1/+ |
| 2014 | 8/170 | 3/++ | 2/++ | 2/++ | 1/+ |
| 2015 | 9/180 | 3/++ | 2/++ | 3/+++ | 1/+ |
| 2016 | 12/144 | 3/++ | 3/++ | 5/+++ | 1/+ |
| 2017 | 12/116 | 3/++ | 3/++ | 5/+++ | 1/+ |
| Quotes—Declaration | " . . . control audit ISO 9001 and 14001 . . . Code of Ethics . . . Platform for company development as a free platform for the discussion with employees . . . "—Čepro presents itself as a very modern and open-minded business. | | | | |
| Comments | Čepro presents the CSR information in a convincing and inter-related manner. | | | | |

**Table 10.** Annual reports of Continental Automotive Czech Republic s.r.o., ID 62024922.

| Year | CSR | R&D | Environm. | Employm. | Others |
|------|-----|-----|-----------|----------|--------|
| 2013 | 2/44 | >1/+ | >1/+ | >1/++ | 0 |
| 2014 | 2/48 | >1/++ | >1/++ | >1/++ | 0 |
| 2015 | 3/48 | >1/++ | >1/++ | 1/++ | 0 |
| 2016 | 3/53 | >1/++ | >1/++ | 1/++ | 0 |
| 2017 | 3/55 | >1/++ | >1/++ | 1/++ | 0 |
| Quotes—Declaration | " . . . CZK 1 441 million on R&D not to be amortized and CZK 3 269 million on R&D to be amortized . . . co-operation with colleges . . . co-operation with R&D center in Ostrava . . . ISO 14001 for environment . . . extensive employee health protection . . . " | | | | |

**Table 11.** Annual reports of Finitrading a.s., ID 61974692.

| Year | CSR | R&D | Environm. | Employm. | Others |
|------|-----|-----|-----------|----------|--------|
| 2013 | 0/34 | 0 | 0 | 0 | 0 |
| 2014 | 0/34 | 0 | 0 | 0 | 0 |
| 2015 | 0/34 | 0 | 0 | 0 | 0 |
| 2016 | 0/34 | 0 | 0 | 0 | 0 |
| 2017 | 0/34 | 0 | 0 | 0 | 0 |
| Quotes—Declaration | " . . . ", i.e., no CSR declarations at all–nothing to be cited. | | | | |

As indicated in Table 8, Hyundai has rather short annual reports which have a set of chapters focusing on CSR (3–9%). Interestingly, along with typical types of CSR information, an impressive set of "other CSR" matters is included, such as various social and cultural projects. Various international standards and national prices are underlined.

As indicated in Table 9, Čepro has a set of short chapters regarding various types of CSR information (4–5%) which, despite their brevity, provide concrete data. Čepro underlines its adherence to international standards, its Code of Ethics and its drive to open communications, i.e., a bottom-up approach involving employees and getting them engaged in company decisions and allowing them to share in the company profits (extra bonuses for employees). Unlike other companies, Čepro seems to link R&D to the CSR.

As indicated in Table 10, Continental has a set of very short chapters entailing various types of CSR information (4–6%) which, despite being brief, provides concrete data and excellent examples. Continental underlines its adherence to international standards and its commitment to co-operate with academia regarding both R&D and hiring new employees. Unlike other companies, Continental seems to link R&D to CSR and even provides data about its large spending on R&D and the appearance and operation of R&D centers.

As indicated in tAbel 11, Finitrading is the only exception among the 10, i.e., this company has filed with the Commercial Register its annual reports with audited financial statements without any CSR information. Finitrading has more than 10,000 employees and is well known in the Czech Republic. Its domain is finitrading.cz and, interestingly, even on it, there is no CSR information provided. As a matter of fact, there is a link channeling all searchers for information and documents from the website of Finitrading on the domain finitrading.cz to the website of the Commercial Registers, i.e., justice.cz. Hence, Finitrading does not provide CSR information at all, which is, considering its size, field of activity (steel and iron) and the large number of employees, surprising, and this fact deserves further exploration and explanations. Nevertheless, this is beyond the scope of this paper.

## 5. Analysis and Discussion

The holistic meta-analysis with respect to the two main purposes of this paper brings together a reflection upon the conceptual and legislative framework and upon their real life operation. The study of the conceptual framework indicates that CSR is a reality of the modern European integration and that CSR reporting belongs to current policies. It is well established that competitiveness resides not only in basic economic performance and outputs, but also in social, environmental, cultural and other elements [13].

The legal framework shows that the EU law has crossed the goal-line with respect to CSR reporting as such, but not yet about its content and arguably even not yet fully about the exact dimensions of this duty [20,28–32]. Generally, it is proposed that the understanding of the extent, format and form of CSR varies considerably [29], across industries [52,62], and on both EU and national levels [63], and that conflicts are not only between various stakeholders but even between various CSR concerns [26]. An apparent call for more law regulations [22] from above is paralleled by a proposition that the bottom-up approach reflecting the current drive of businesses for transparency and for better disclosure is more suitable [64].

A similar hesitation and exchange of opinions can be observed regarding the publication of such reports, namely the use of BRIS, as reflected by very different approaches of the EU member states [27]. Indeed, national particularities have the potential to clarify and strengthen the content and form of the CSR reporting duty and its e-availability, but in the case of the Czech Republic, similarly to the majority of other EU member states, this potential is only partially realized. Namely, with but little exaggeration it can be argued that the Czech drive for the form and accessibility [27] outdistances concerns over the content [28], i.e., the legal duty in the Czech context demands basic CSR information to be present in the freely e-available annual reports of certain subjects.

Indeed, the conceptual and legislative framework with respect to the Czech context implies only one strict, hard and rather general conclusion with respect to the largest Czech companies—they have the legal duty to prepare annual reports and include in them CSR information. Period. This opens a large window of opportunity for companies in their decision making processes on and about the included and ultimately published CSR information.

However, the recipients of this information, i.e., the audience for reading these annual reports with CSR information, is very heterogeneous and does not want to have "some" information. In order to avoid futility, a reasonable quantity and quality of data about R&D, environmental protection and employment matters is to be provided in a searcher-friendly manner, because this information is expected by a large and heterogeneous pool of subjects [65]. Some of these "searchers" are current or potential business partners, and if the company wants to be perceived as a reliable fit for them, it had better convey such a message and endorsement of matching concepts [66], e.g., by including well developed CSR information in their freely electronically available annual reports. Others of these "searchers" are consumers and the general public, i.e., the large section of the public that is concerned about CSR issues and, at the same time, having neither the time, nor the desire, nor the capacity for intensive information gathering of CSR material [67]. Naturally, even employees can become "searchers" and the CSR information can have a positive impact on their filing vis-à-vis to their employer and on their readiness to perform their job duties [68,69]. In sum, regardless of whether they are outsiders or insiders, there is an increasing demand for better and more accessible information on CSR [12,68,69] and the public draws its own conclusions.

For example, one previously published micro-study of CSR importance with respect to large Czech companies revealed that two-thirds of consumers paid attention to the CSR information provided and even became influenced by it [70]. However, other studies indicate that the majority of companies do not file, or try to avoid having to file, annual reports with CSR information with the Commercial Register [28] and so miss out on the opportunity to provide a public declaration to society about their CSR commitment. Since it is well established that business ethics is regarded as a significant factor with an impact on both profitability and corporate image in the EU, and in particular in a Central European context [8–10], the empirical observation might suggest that this is generally underestimated by Czech companies.

Moving to the sample used for the Czech case study, consisting of the 10 largest Czech companies by revenue, the EU law analysis reveals that they are subject to the legal duty to provide both financial and non-financial statements, i.e., they have to include CSR information in their reports. The analysis of the Czech law added to this manner of content command a very important and not so common in other EU member states, legal duty about the form, or even more specifically about the form's presentation. Namely, Czech companies of all sizes have to prepare annual reports and they have to file them with the Commercial Register to be e-published and so made freely available to the public-at-large without any restrictions. By the operation of EU law, this data is further, at least partially, migrated to the EU central system BRIS on the e-Justice Portal.

This performed case study of annual reports brings forth a pioneering and very useful insight regarding types, quantity, quality and even the declaration dimension of the realization of the legal duty discussed. Table 12 extracts and presents an overview of information collected from all these companies and all their annual reports.

**Table 12.** Overview—CSR information in annual reports of the 10 largest Czech companies in 2013–2017.

|  | Extent—Quantity | Depth—Quality | Declaration Potential |
|---|---|---|---|
| CSR | 0–10% of annual report | Generally + or ++ | differences in the use |
| R&D | 0–2 pages | + | Underemployed |
| Environm. | 1–4 pages | ++ | Employed |
| Employ. | 1–5 pages | +++ | strongly employed |
| Others | 0–3 pages | + | Employed |
| Comments | 9 out of 10 companies included CSR information in their annual reports, the CSR information took a similar share of the annual report and no dramatic changes occurred between 2013 and 2017. Generally, companies include CSR information to a similar extent, but in a different manner and depth. Certain companies have CSR chapters in their annual reports while others spread the CSR information into different segments of annual reports, often even without treating it as CSR information. References to international standards, official prices and awards were mentioned, occasionally misleading information was presented. | | |

The above table suggests a wide spectrum with respect to the type, quantity and quality of the CSR information provided by the largest Czech companies. This spectrum is provided freely and online to the public-at-large, from partners and competitors over to state authorities and even to consumers. Indeed, the 10 largest Czech companies are subject to this legal duty, are aware about it and, except for one (Finitrading), satisfy it. They prepare their annual reports, or have an outside firm prepare them, have them verified and audited, and e-file them with the Commercial Register. These annual reports are truly digitally available without any restrictions and they include CSR information. However, neither EU law nor Czech law provides mandatory and detailed instructions about the type, extent and depth of this information. Namely, the legal duty is set up in a rather general and vague manner. This leaves a rather large discretionary space in its perception and satisfaction, and it is up to these companies how strongly they will adhere to the CSR concept and how extensively they will report about it via annual reports. Boldly, it is up to these companies regarding how much and to what extent they will provide CSR information about their R&D, environmental, employment and other matters in their annual reports or somewhere else, e.g., by making statements on their internet pages. The manner in how they proceed, regardless whether via annual reports or otherwise, reflects their perception of the CSR, as well as their readiness to use it as a public declaration.

Since the case study considered the period of five years and investigated all annual reports filed by these companies, it is sufficiently indicative, at least regarding the most official and formal communication pathway for the CSR declaration. The yielded results confirmed that these companies do satisfy their legal duty, i.e., they meet the mandatory threshold and include in their annual reports the CSR information and file these reports with the Czech Commercial Register. The companies act in a predictable and stable manner and so no dramatic differences develop between the annual reports for the same company during the 2013–2017 time frame. However, the analysis of the CSR information provided in these annual reports revealed that certain companies provide CSR in a much more reader- or consumer-friendly manner and are well organized, while other companies spread the CSR information through their annual reports, often without any logic. Although the share of CSR information in the annual reports is fairly similar among the observed companies (1–10%), important differences in the quality of the information exist. Often much better data is provided regarding employment matters and possibly as well environmental protection than regarding social matters and R&D, which is occasionally not even considered as related to CSR. The top focus on the employment matters is a product of several factors, such as the impact of economic downturns on structural unemployment [71], a generally large social focus on job safety and the current economic situation in the Czech Republic.

Interestingly, on the one hand, the drive to provide robust statements about CSR with respect to employment matters sometimes will lead companies into presenting misleading information and to presenting the CSR as a 'bonus', even though it is something required by mandatory labor law provisions (e.g., the statement about the observance of the labor law). On the other hand, some companies are determined to provide truly objective and correct CSR information. Thus they proudly point out some of their objective achievements, prizes and awards received and officially followed standards. The teleological and purposive approach to annual reports reveals that, for some Czech companies, typically from the oil-gas-chemical industries, the CSR information in the annual report is a great vehicle for a public declaration to society, while for other companies is it a mere duty.

Highly inspiring is the holistic projection of the data yield in the EU and global comparative context. Namely, these 10 Czech Companies and their 50 annual reports mirror trends, phenomena and particularities of the CSR and CSR reporting in annual reports in other jurisdictions in and even outside of the EU. Indeed, especially in a central European context, similar patterns can be observed. For example Slovak, Polish, Hungarian, Austrian and Italian companies address their CSR reporting duty formalistically [10,29,48,72] and often without engagement in depth and thus behave similarly to Agrofert and Hyundai. While Polish companies such as Agrofert and Hyundai are not prompted to state ethical considerations, Hungarian companies seemed more committed to ethical concerns and

ethical codes, as do Čepro, Foxconn or Unipetrol [10]. Further, it has been established in Spain that the CSR and CSR reporting is influenced by the appointment process of directors, i.e., companies with directors appointed by controlling shareholders more firmly embrace CSR strategies and provide better CSR reporting than do companies with directors appointed by funds and funds representatives [73] and this can be observed in the case of Škoda and Čepro. In addition, Spanish companies were used to demonstrate that internal and external CSR contributes to the enhancement of the product and service quality and should be perceived and reported as such [74] and this can be observed as well in the case of Škoda. Regarding French companies and their CSR reporting, there has been observed a strong link between the CSR and branding, in particular that the store brand with a price-quality ration contributes to the development of a positive price image and CSR image [75], exactly as in the case of ČEZ or Škoda. Indeed, outside the EU, especially in the USA, South Korea and Japan, a very interesting relation is observed between the CSR drive and commitment, the use of luxury brands and powers of individuals [76] and undoubtedly Škoda and Hyundai would like to adhere to it. Nevertheless, strong voices presenting the CSR commitment as a win-win situation for shareholders and stakeholders [77] from the other side of the Atlantic, and matched by EU high expectations and only soft, not binding, guidelines [78,79], are not (yet) echoed by the Czech reality and annual reports of (at least the selected) Czech companies.

## 6. Conclusions

This paper and the underlying study on its two main purposes contributes to knowledge and discussions going to the very roots of current interaction and behavior in the market and even beyond. Every human community, including the global business community, needs the preservation of a set of orders under the auspices of certain values and this while working towards the common good [80]. Over the last few decades, CSR has grown to become one of the key concepts intimately linked to business ethics and even beyond, in short, the general direction for the future [40]. In the EU, CSR is recognized and supported both by legal provisions generated by the EU and EU member states and by the actions of all stakeholders, especially companies. Nevertheless, this is merely a general framework presentation and there are many questions and issues linked to the determination of the exact dimension of the mandatory features of the legal duty to provide CSR information, and in particular in annual reports and the e-publication of these annual reports. The previously performed national case studies revealed the national law particularities with respect to CSR reporting and its problematic effectiveness and efficiency, e.g., despite the apparent compliance with regulatory demands and the growing number of sentences provided, no substantial improvement of completeness of CSR reporting occurred in Italy [72].

The 10 largest Czech companies are subjects of the legal duty to prepare and e-file with the Commercial Register their annual reports with CSR information, but this legal duty is set up in a rather general and vague manner. This leaves open a rather large discretionary space in its perception and satisfaction. Boldly, it is up to these companies how much and in what depth they will provide CSR information about their R&D, environmental, employment and other matters in their annual reports. The manner in which they proceed reflects their perception of the CSR and their readiness to use it as a public declaration. The performed case study considers a period of five years and investigated all annual reports filed by these companies. The results confirmed that these companies generally do satisfy their legal duty, i.e., 90% of them meet the mandatory threshold and include in their annual reports CSR information and file these reports with the Czech Commercial Register. However, the analysis of the CSR information provided in these annual reports revealed that the manner of CSR reporting is not settled. Perhaps the only three common features are that generally (i) the CSR information usually appears on only 1 to 10 pages, or even does not appear at all; (ii) represents 1–10% of the total number of pages of the annual report and (iii) touches most heavily on employment matters. Otherwise, significant differences exist. Regarding the form, some annual reports are well organized with respect to CSR, while others are not, some of them have a special CSR chapter while

others spread the CSR information across the entire annual report. Regarding the types, the importance and developments vary and sometimes R&D is totally skipped. Regarding quality and concreteness, there is very little in common, where some companies can, on one or two pages, provide detailed and convincing CSR information while other companies have their pages of the annual report filed with general and unconvincing CSR information. For sure, each annual report has the potential to convey a declaration to society, but this potential is not always developed. In addition, these declarations have in common only the adherence to international standardization. Otherwise, each company seems to exercise quite a bit of discretion about how they used the annual report and CSR information for information and marketing purposes. Furthermore, it can be proposed that these trends described via the observed Czech companies reflect, to a certain extent, patterns observed abroad, especially with respect to Central Europe.

There are some limitations attached to this research and it would be very instructive and useful to address them in the near future. First off, the case study entailed annual reports of the 10 largest Czech companies filed for 2013–2017 and naturally expanding this sample would be extremely useful and would boost its significance. This expansion should entail various national aspects (not only Czech companies, e.g., contrast it to Italian companies), the quantitative aspect (more than 10 companies) and the company's size aspect (adding smaller companies). This would be both very interesting and feasible, in short, no small feat, due to the (un)availability of data and reports from other jurisdictions or regarding smaller businesses and due to the complexity of working with many hundreds of reports, and these in different languages. This would make them very difficult to compare, to say nothing of the numerous pitfalls that would arise due to their differing linguistic aspects. Secondly, it would be interesting to go into more depth in the annual reports and deeply discuss the manner of their approach and inclusion of the CSR information (or the lack of such an inclusion). Thirdly, it would be enlightening to study these companies in more depth and about other channels, including their various presentations dealing with CSR. Fourthly, it would be inspiring to focus even more on the interplay of the CSR and ethics with the business results of these companies. Finally, the objectivity and richness of perspectives would be improved by enrolling a group of experts to process the yielded data and to rank and comment on the CSR information provided, i.e., to move from the group of three experts following basic guidelines to a large group of experts following more sophisticated score cards and ultimately going for synthetic qualitative data processing.

CSR is a much needed concept in the second decade of the 21st century, throughout the entire global setting and if in the EU, member states and even Europeans are serious about the proclaimed smart, sustainable and inclusive growth, they have to not only work both hard and in compliance with CSR and related business ethics demands, but also inform and be informed about it. In the digital post-modern society, increasingly, knowledge is power and information is the top commodity. The enhancement of awareness and open information about the CSR are needed and can contribute to a competitive advantage. The inclusion of genuine and appropriate CSR information in freely available digital annual reports is not a bureaucratic duty set out by law, instead it is a great opportunity offered by our digital era for all stakeholders.

**Funding:** This research and resulting contribution were funded by Czech Science Foundation, grant number GA ČR No. 17-11867S "Comparison of the interaction between the law against unfair competition and intellectual property law, and its consequences in the central European context."

**Conflicts of Interest:** The author declares no conflict of interest.

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
