# Peer review of "Corporate Social Responsibility Information in Annual Reports in the EU—A Czech Case Study"

_sustainability, doi:10.3390/su11010237_

Round 1

Reviewer 1 Report

The topic is up to date and interesting.

In general, the manuscript is informative and well written. Some of the strengths I would like to highlight are the following ones.

·         Interesting research topic

·         Appropriate literature review

·     Appropriate structure of the manuscript in line with the general rules for research papers (introduction, framework, etc.).  

Nevertheless, the manuscript has important weaknesses that have to be solve. Here are the most important ones.

Objective. The author state a double goal (page 1, line 13-14). Regarding the first part of it, “identifying the parameters of this annual reporting duty (about RSC) and…” I am not sure about if this research comply with this objective. After reading the manuscript, this issue is not clear at all. Maybe this part of the objective should be redefined or it should be advisable complement the framework in order to comply with it.

Methodology

· The description of the methodology used is not very clear, especially the first part referring to the review of the literature. It talks about the teleological interpretation of legislation, the use of meta-analysis, search in database such as Scopus and WOS, ... but nothing remains clear how it was carried out or how everything is interrelated.· There is a problem of subjectivity related to the qualitative analysis of the information provided by the companies of the sample. The author said that himself (expert in these subjects) was responsible for carrying out this information assessment, but it would be much more robust if a group of experts (got several opinions) had been used to carry out this assessment.· Regarding the representativeness of the sample, I am not sure if the selected companies (the 10 largest Czech companies) are very representative of the total of Czech companies. How can this representativeness been justified? In fact, the author already talks about this issue (problem) in conclusion section, as a limitation of the research. However, it would be advisable that the study include not only large companies but also SMSEs ones, instead of being something to be addressed in the near future, in order to improve the study.

Case Study. It must be better justify why the study focuses on Czech companies and not on companies from others countries. It is not very clear if it is because it is a convenience sample, if it is because the Czech companies have some particularity that can make it easier to generalize the results to other countries, etc.

Results. Some information should be relocated. In results section (page 6, lines 290-317). The author recapitulate about the research gap, the state of the art, justify why this study is carried out and finally describe the case study. In my opinion, this is repetitive information that have been summarized and relocated at the end of section 2. Results section should be used only for present the main results of the study.

Author Response

Responses to Reviewer 1, 2 and 3 - Comments – FYI, all changed highlighted in respective colours

Reviewer 1

Point 1: Objective. The author state a double goal (page 1, line 13-14). Regarding the first part of it, “identifying the parameters of this annual reporting duty (about RSC) and…” I am not sure about if this research comply with this objective. After reading the manuscript, this issue is not clear at all. Maybe this part of the objective should be redefined or it should be advisable complement the framework in order to comply with it.

Response 1: The reviewer is correct; the first part was addressed in the prior version in a short and perhaps not completely clear manner. Therefore, the new version is amended (amendments are highlighted in yellow) and the parameters of the legal duty regarding the CSR reporting are described in a more clear and developed manner.

Point 2: Methodolgy. The is a problem of subjectivity related to the qualitative analysis of the information provided by the companies of the sample. The author said that himself (expert in these subjects) was responsible for carrying out this information assessment, but it would be much more robust if a group of experts (got several opinions) had been used to carry out this assessment.· 

Response 2: Yes, a group of experts would generate more robust and objective assessments and the synthetic approach would boost this further. Nevertheless, due to the extent of the paper and the availability resource, the author moved to a compromise solution – she asked two colleagues, specialists with expertise regarding the corporate reporting (EDC, LM), prepared the ranking guidelines (for what + or what ++ and for what +++) and each moved on to go over these 50 annual reports. The use of 3 experts is newly mentioned as a limitation of the paper (amendments are highlighted green).

Point 3: Methodology Regarding the representativeness of the sample, I am not sure if the selected 

companies (the 10 largest Czech companies) are very representative of the total of Czech companies. How can this representativeness been justified? In fact, the author already talks about this issue (problem) in conclusion section, as a limitation of the research. However, it would be advisable that the study include not only large companies but also SMSEs ones, instead of being something to be addressed in the near future, in order to improve the study. It must be better justify why the study focuses on Czech companies and not on companies from others countries. It is not very clear if it is because it is a convenience sample, if it is because the Czech companies have some particularity that can make it easier to generalize the results to other countries, etc.

Response 3: Yes, it would be definitely more interesting and illustrative. Nevertheless, due to the extent of the paper and the available resources, and as well considering the two purposes, the author intentionally selected subjects satisfying the legal parameters criteria (500 employees and sro or as form) and at the same time being “well known”. Hence the pick of the “biggest” 10 companies led to the sample consisting of companies which basically all had filed their reports (potentially with CSR information). In addition, these ten companies are known to the public-at-large (including foreigners) and everybody can check the information about them provided in the paper (go to justice.cz) or even test it and expand it further (go to www pages of these companies).

In sum, I understand that the selection of the sample has to be better explained and so I did not change the sample, but following this comment of the reviewer, I added an explanation in the paper (amendment in turquoise).

Point 4: Results. Some information should be relocated. In results section (page 6, lines 290-317). The author recapitulate about the research gap, the state of the art, justify why this study is carried out and finally describe the case study. In my opinion, this is repetitive information that have been summarized and relocated at the end of section 2. Results section should be used only for present the main results of the study.

Response 4: Yes, this is right and I have made the relocation as suggested and corrected the results sections (amendments in grey)

Reviewer 2

Point 5: CONCEPTUAL FRAMEWORK

In this section the author explores the role of sustainability in legislative domain. It misses to consider the strategic role of companies’ disclosure, not just the compliance issues related to CSR. Indeed, CSR information in the annual report is often related to companyies’ proactivity on CSR and it’s used as way to communicate their efforts besides regulatory stringency. How do you consider that?

Response 5: Yes, a very good point, information was added in Part 2 (Amendments highlighted in red)

Point 6: CONCEPTUAL FRAMEWORK

Indeed, this table/figure should show the different information required classified according the different regulatory source.

Response 6: Yes, the legal framework needs to be clearly explained. As a matter of fact, this was expressed in point 1 by Reviewer 1 and I have made changes and added explanations plus summary regarding legal parameters (amendments are highlighted in yellow). I consider this sufficient. As well, I am afraid that adding an additional large table would make the paper too long and less compact. However, if the Reviewer 2 is still convinced that despite the changes made an additional table is needed, then please provide me with such information and I will prepare and add such a table.

Point 7 (Reviewer 1 and Reviewer 2 raised the same point): METHODOLOGY

The main weak  point regards how they ranked the informationl. The value assigned by “legal expert” is not enough. How does sujectivity and personal prejudice of the expert can be avoid? At least three different expert shoul rank the sentence and, in case of agreement, consider that value, otherwise go on with other point of view. Otherwise, the author can develop a syntetic measure of CSR information quality, for instance by carrying on a content analysis on the text.Thus, a more rigourous method should be to increase research reliability.

Response 7: Yes, here both reviewers raised the same points (see point 2 and point 3 of the Reviewer 1 and point 7 of the Reviewer 2) and I fully agree, i.e. I have changed my paper accordingly, please see my explanation to point 2 and point 3 (amendments in light green and turquoise).

Point 8: METHODOLOGY

Moreover, author take into account four dimensions of CSR (R&D, ENVIRONMENT, EMPLOYEES AND OTHERS) that are not well described into theoretical section.In order to use these dimensions to evaluate the level of quality of CSR information, authors should develop theoretical speculation on such dimension of CSR. In particular, he/she should specifically identify the level of regulatory pressure on each specific dimension of CSR.  

Response 8: Yes, this is right, I added the information (the amendments are highlighted in purple/violet)

Point 9: RESULTS

The author provide a table for each company. Results should be better explained and presented.

Konec formuláře

Response 9: Yes, the review of 50 annual reports generated many interesting points worthy of presentation in tables with comments. Due to the size of the paper and the drive for compactness, I have added one row-line to each table with comments (amendments highlighted in brown), but if  Reviewer 2 still thinks that this is insufficient and more tables are needed, then please inform me and I am open to expand even more these tables and possibly add other table(s).

Reviewer 3

Point 10: AIM OF THE PAPER. The aim of the paper is not clear. In different parts of the papers, different aims are presented….. Again, I do not see how the first purpose is addressed in the paper  and I do not understand the "why should we care question" in the second research question (please see below)

Response 10: Yes, this is very similar to the point 1 raised by the Reviewer 1 and led to the changes (amendments are highlighted in yellow).

Point 11: CONTEXT/CASE STUDY. I understand that the data that you have comes from Czech Republic, however this is not enough to justify the importance of the research. I kindly invite you to reply to "why should we care question" in the introduction and clearly explain why this study could be important in the EU context. A brief analysis of previous studies that have analysed the adoption of the EU directive in different EU countries is needed. See for instance the following papers (Costa and Agostini, 2016, Social and Environmental Accountability Journal; Meditari special issue 2019 http://www.emeraldgrouppublishing.com/products/journals/call_for_papers.htm?id=7942, which also provides useful references)

Response 11: Yes, I have provided additional information about the focus on the Czech jurisdiction and contrasted it with the correctly proposed Italian study, and underlined that this has a potential for further research (amendments are highlighted in dark green, including reference to Costa Agostini).

Point 12: METHODS AND METHODLOGIES. The paper is very weak in terms of methodology. First of all I do not see the quantitative and qualitative part of the analysis. …., why number of pages as metrics of analysis? most studies suggest "sentences" as the most reliable unit of analysis; how is quality assessed?

 Response 12: Yes, the methodology was a subject of questions of all three reviewers and that part has been amended, including the explanation of the selected „page“ criterion (amendments are highlighted in light green and turquoise).

Conclusions – I have addressed all points raised by reviewers. I am grateful for these points, because they were relevant, I have changed my paper accordingly and the current version represents a true improvement. However, I admit that reviewers might expect even more dramatic changes (especially adding more information and more table), but at this point I am afraid to go for it, because this might make my paper lose its momentum and became far too lengthy.

Reviewer 2 Report

The overall aim of the paper is interesting withoun any doubt. However, the paper shows some point of weakness, expecially in the methodological section, that need to be improved to be published.

CONCEPTUAL FRAMEWORK

In this section the author explores the role of sustainability in legislative domain. It misses to consider the strategic role of companies’ disclosure, not just the compliance issues related to CSR. Indeed, CSR information in the annual report is often related to companyies’ proactivity on CSR and it’s used as way to communicate their efforts besides regulatory stringency. How do you consider that?

Moreover, in order to give to reader a better understanding of the different level of legislative framework and regulatory request on CSR practices, a final table or figure is needed.

Indded, this table/figure should show the different information required classified according the different regulatory source.

METHODOLOGY

Author employs a meta-analysis to aswer to the main research question.

The main weak  point regards how they ranked the informationl. The value assigned by “legal expert” is not enough. How does sujectivity and personal prejudice of the expert can be avoid?

At least three different expert shoul rank the sentence and, in case of agreement, consider that value, otherwise go on with other point of view.

Otherwise, the author can develop a syntetic measure of CSR information quality, for instance by carrying on a content analysis on the text.

Thus, a more rigourous method should be to increase research reliability.

Moreover, author take into account four dimensions of CSR (R&D, ENVIRONMENT, EMPLOYEES AND OTHERS) that are not well described into theoretical section.

In order to use these dimensions to evaluate the level of quality of CSR information, authors should develop theoretical speculation on such dimension of CSR. In particular, he/she should specifically identify the level of regulatory pressure on each specific dimension of CSR.  

Again, the different variable considered in the study can be summarized in a table.

 RESULTS

The author provide a table for each company. Results should be better explained and presented

Author Response

(The authors gave the same response as above.)

Reviewer 3 Report

Dear Author,

thank you very much for the opportunity to read your paper. I found the manuscript quite interesting, indeed it addresses a relevant topic related to CSR disclosure in the European annual report. It explores the case of Czech Republic, without adequately positioning it the broader European debate. My major concern related to the focus of the paper (and its declared purposes) and the methodology (which needs to be strongly improved). I express my detailed concerns below:

1.     AIM OF THE PAPER. The aim of the paper is not clear. In different parts of the papers, different aims are presented. For instance, the first purpose is to determine the EU framework and the second is to assess its real life application, such as explained in the Introduction. I have major concern regarding the first purpose, since it is more related to building and contextualise the study. Moreover, on page 5 regarding "Materials and Methods" the paper declared two different purposes, i.e. i) to identify the parameters of the annual reporting legal duty with respect to CSR and ii) to check the realization by Czech companies. Again, I do not see how the first purpose is addressed in the paper  and I do not understand the "why should we care question" in the second research question (please see below)

2. CONTEXT/CASE STUDY. I understand that the data that you have comes from Czech Republic, however this is not enough to justify the importance of the research. I kindly invite you to reply to "why should we care question" in the introduction and clearly explain why this study could be important in the EU context. A brief analysis of previous studies that have analysed the adoption of the EU directive in different EU countries is needed. See for instance the following papers (Costa and Agostini, 2016, Social and Environmental Accountability Journal; Meditari special issue 2019 http://www.emeraldgrouppublishing.com/products/journals/call_for_papers.htm?id=7942, which also provides useful references)

3. METHODS AND METHODLOGIES. The paper is very weak in terms of methodology. First of all I do not see the quantitative and qualitative part of the analysis. The paper needs to i) better explain its epistemic approach (really confused in the current version), ii) describe the content analysis developed (I do not see the methodological choices at the bases of the paper, why number of pages as metrics of analysis? most studies suggest "sentences" as the most reliable unit of analysis; how is quality assessed? The paper does not mention it; iii) what does "Holistic meta analysis" mean? I do not see the holistic part, as I do not see the qualitative dimension of the study. To me the database (50 annual reports) is too weak for developing a proper quantitative analysis, and at the same time no complementary data are actually presented in order to complement the analysis in a qualitative way. No interviews, no questionnaires are introduced. I strongly recommend the author(s) to better explain their methodologies and to explain the data set in a quantitative (more annual report) or qualitative (Interviews) way.

Some references on content analysis for CSR reports:

Useful References:

Hooks, J. and van Staden, C.J. (2011), “Evaluating environmental disclosures: The relationship between quality and extent measures”, The British Accounting Review, Vol. 43 No. 3, pp. 200–213.

Milne, M.J. and Adler, R.W. (1999), “Exploring the reliability of social and environmental disclosures content analysis”, Accounting, Auditing & Accountability Journal, Vol. 12 No. 2, pp. 237-56.

Pesci & Costa (2014) Content Analysis of Social and

Environmental Reports of Italian Cooperative Banks: Methodological Issues, Social and

Environmental Accountability Journal, 34:3, 157-171, DOI: 10.1080/0969160X.2014.904239

Unerman, J. (2000), “Methodological issues - Reflections on quantification in corporate social reporting content analysis”, Accounting, Auditing & Accountability Journal, Vol. 13 No. 5, pp. 667-681.

I understand the suggestion provided are challenging, but I would really encourage the author to take them seriously in order to make the paper publishable. 

Author Response

(The authors gave the same response as above.)

Round 2

Reviewer 1 Report

Dear author,

Thank you, so much, for your responses. Please, see below my comments and suggestions about them.

Point 1 – Objective

Although the author has tried to explain in a more detailed what is the main (double) purpose of the paper, I am not sure if it is clear enough and if the manuscript really complies with it.  

The author states that de double purpose of the paper is “determine this framework (the status quo of the CSR reporting in the EU) and to assess its real life application” (lines 39-40). However, it is not the purpose of the present study. This is a medium-large term goal and, specifically, in this paper (an exploratory research), the main objective is analysing the case of the Czech Republic in order to stablish a starting point to analyse what happens in EU context.

For that reason, it would be advisable to better link the idea states in lines 39 -40 with the idea states in lines 40 – 43.

Point 2 – Methodology

In my opinion, the problem of the subjectivity still persists. Although three experts has been used in order to carry out the assessment of the information on CSR, it is important to note that one of these three experts is the author of this paper, who has designed the study, carried out the data collection, etc. So I am wondering if it would be considered as a potential bias problem.

Additionally, regarding the measurement scales, the author explains that these three experts used universal guidelines (lines 341-350). However, it is not clear how these guidelines (“measurement scales”) were constructed or who developed them. Was the author of this paper who developed these scales, although it is one of the experts who asses the information on CSR? Did other authors (studies) previously develop these guidelines? How can one know if these guidelines are suitable, (reliable) to assess the information collected?

It is true that the author argues that the research should be further developed by enrolling a group of experts. Moreover, he/she highlights the necessity of using a more formal instrument of assessment (lines 626-630). I consider that these are not limitations to comply with in a near future. On the contrary, I think that it is something to do now in order to improve the methodology and solve the subjectivity problem. I highly recommend the use of the Delphi method.

Point 3 – Methodology

Regarding the representativeness of the sample (which is make up of only big companies), taking into account the changes done and explanations given by the author, now it is more obvious that it is a convenience sample and it would be considered that it has been “appropriately” justified.

However, in spite of the changes made, I consider that it has not been adequately justified the fact that the study is focused on Czech companies instead of doing it on companies from other countries. In this sense, the author states, as a limitation (lines 618-619), the difficulty of analysing information from other countries because of the language. However, most of the companies from EU countries give the information in English although it is not their mother tongue.  In my opinion, it is not a powerful argument for not including other countries in the research. In addition, I am still not able to see the contribution to this paper (or its importance) in the EU because, as the author states, the legislation on CSR is different in each member country of the EU. I think this is a point that the author has to improve, trying to highlight the contribution of his/her study. I mean, how the fact that the study is focused on Czech companies can contribute in the field of the CSR.

Point 4 – Results

When the author relocates some of the paragraphs from the results section to other sections, he/she forgot to (re)number the references. For example, references [42, 57] in line 114 or references [58-62] in line 116 must not appear before references [37], [38], etc. Please, check them and read the manuscript guidelines for authors.

In addition, I still think that some of the information from the results section (lines 364-384) should be relocated as well, or it would be simply removed, because it is not directed linked with the results. It these paragraphs the author sums up again the information previously stated in other sections of the manuscript and, in my opinion, it is not necessary to duplicated it at all.

Author Response

Sustainability 394041

Corporate Social Responsibility Information in Annual Reports in the EU – A Czech Case Study

Responses to Reviewer 1, 2 and 3 - Comments – All previous changes highlighted in yellow, all new changes hightlighted in green.

Reviewer 1

Point 1: Objective

The author states that de double purpose of the paper is “determine this framework (the status quo of the CSR reporting in the EU) and to assess its real life application” (lines 39-40). However, it is not the purpose of the present study. This is a medium-large term goal and, specifically, in this paper (an exploratory research), the main objective is analysing the case of the Czech Republic in order to stablish a starting point to analyse what happens in EU context.

For that reason, it would be advisable to better link the idea states in lines 39 -40 with the idea states in lines 40 – 43.

Response:

Sure, I have modified it to make clear that I want to (i) research and indentify the legal framework, i.e. the exact extent of the current and applicable legal duty covering the CSR reporting in the Czech Republic (this is stated by the EU law and Czech law) and to (ii) assess how this legal duty is observed in real life. The new wording is as follows:

This leads to the need for having two main and highly important purposes for this paper – to determine this EU and Czech legal framework and to assess its real life application. Namely, an understanding of the current exact parameters of the CSR reporting in a selected jurisdiction within the EU, such as the Czech one, needs to be established first, so as to make a foundation for the performance of a national case study regarding the reality of the CSR reporting in such a jurisdiction.

Point 2: Methodology

In my opinion, the problem of the subjectivity still persists. Although three experts has been used in order to carry out the assessment of the information on CSR, it is important to note that one of these three experts is the author of this paper, who has designed the study, carried out the data collection, etc. So I am wondering if it would be considered as a potential bias problem.

Additionally, regarding the measurement scales, the author explains that these three experts used universal guidelines (lines 341-350). However, it is not clear how these guidelines (“measurement scales”) were constructed or who developed them. Was the author of this paper who developed these scales, although it is one of the experts who asses the information on CSR? Did other authors (studies) previously develop these guidelines? How can one know if these guidelines are suitable, (reliable) to assess the information collected?

It is true that the author argues that the research should be further developed by enrolling a group of experts. Moreover, he/she highlights the necessity of using a more formal instrument of assessment (lines 626-630). I consider that these are not limitations to comply with in a near future. On the contrary, I think that it is something to do now in order to improve the methodology and solve the subjectivity problem. I highly recommend the use of the Delphi method.

Response:

This is an excellent point and I have completely reconsidered and redone this part, i.e. I have embraced the Delphi method, have improved the guidelines and prepared questionnaires and sent them to the previous two experts along with additional one expert (two woman with economic and law background and one man with economic background, all having experience with annual reports and mastering Czech and English and having corresponding college degree and at least 20 of executive job experience). Hence the pool of experts included 3 members and I was not one of them. I processed the feedback from the first round and prepared a summary which I communicated to them. Thereafter, they made few changes with respect to their prior answers. The data resulting from the second round was used for the paper.

Unfortunately, considering the available resources and the support provided by my grant as well as taking into account the time restrictions and the need to have the poll of pre-defined experts, a further  extension of the group of the experts (e.g. from 3 to 5) is not feasibility.

The new wording is as follows:

. The quantitative criterion of pages rather than sentences was selected due to the linguistic, especially stylistic and pragmatic semiotic, particularities of the Czech language belonging to the Slavic language group. The qualitative aspect was addressed by the holistically manual approach employing a simplified Delphi method. Namely, each and every one of these 50 annual reports was carefully read through by three experts on corporate matters including reporting (EDC, LM and RKM, i.e. none of these three experts was the author of this article) while following a universal set of guidelines and simple questionnaires prepared by the author. All three experts master both Czech and English, have college degrees, experience with annual reporting, at least 20 years of executive job experience and a strong law and/or economic background. Two of them are women and one is a man. Hence, their replies met the expertise expectations. These 1st round replies were processed by the author and, based on them, the author prepared a summary which was communicated to these three experts for the 2nd round. Thereafter, they made a few changes with respect to their prior answers and sent their updated replies to the author. This data, resulting from the 2nd round, was used for the paper.

Specifically, based on these guidelines and questionnaires, each of these three experts categorized the provided CSR information (+) or (++) or (+++). The guidelines required ranking as no more than general information (+) all universal and proclamation-type statements lacking a relationship to real and controllable actions or omissions; to ranked as more developed and concrete information (++) all statements leading to a single real and controllable action or omission or participating on general CSR trends; and as robust information (+++) all statements about real and controllable actions culminating in an exemplary CSR behavior linked to the particular business and which was made public and regarding the existence of which is beyond any doubt. Hence, plain statements such as “we recognize the importance of environmental protection” were (+), more developed statements such as “we make our products in an environmental friendly manner by using this and not that” were  (++) and information about pro-active tangible CSR behavior, such as “although  we provide services and do not directly pollute the environment, we decided to revitalize the park by the daycare XY by planting 100 trees and by being responsible for the ongoing care for them … and because of this and other acts, we received an award …. (or see photos of this park below)” were (+++). Hence, each of these 50 annual reports has been seen and ranked by three experts independently. The results from the 1st round was processed by the author and resent to experts which provided adjusted results in the 2nd round. These results were compared and, in the case of still different results (one expert giving more or less ++ than others), then these three experts conferred with the author and together agreed about the proper ranking. In addition, the analyses performed by these three experts, while studying these reports and later confronting their ranking, allowed for extracting critical statements, quotes and declarations to be demonstratively indicated in the tables below.

Point 3:

However, in spite of the changes made, I consider that it has not been adequately justified the fact that the study is focused on Czech companies instead of doing it on companies from other countries. In this sense, the author states, as a limitation (lines 618-619), the difficulty of analysing information from other countries because of the language. However, most of the companies from EU countries give the information in English although it is not their mother tongue.  In my opinion, it is not a powerful argument for not including other countries in the research. In addition, I am still not able to see the contribution to this paper (or its importance) in the EU because, as the author states, the legislation on CSR is different in each member country of the EU. I think this is a point that the author has to improve, trying to highlight the contribution of his/her study. I mean, how the fact that the study is focused on Czech companies can contribute in the field of the CSR.

Response:

This is a stimulating and challenging proposition and I have put efforts to make significant improvements in this respect, i.e. I have modified n excellent point and I have completely reconsidered and redone this part, i.e. I have expanded the paper and brought a comparative demonstration showing how the information generated from the case study matches with status quo observed abroad. Naturally, I have added appropriate references.

The new wording is as follows:

Highly inspiring is the holistic projection of the data yield in the EU and global comparative context. Namely, the ten Czech companies and their 50 annual reports mirror trends, phenomena and particularities of the CSR and CSR reporting in annual reports in other jurisdictions in and even outside of the EU. Indeed, especially in the central European context similar patterns can be observed. For example Slovak, Polish, Hungarian, Austrian and Italian companies address their CSR reporting duty formalistically [10, 29, 48, 72] and often without engagement in the depth and thus behave similarly to Agrofert and Hyundai. While Polish companies as Agrofert, Hyundai are not prompted to state ethical consideration, Hungarian companies seemed more committed to ethical concerns and ethical codes as Čepro, Foxconn or Unipetrol [10]. Further, it has been established in Spain that the CSR and CSR reporting is influenced by the appointment process of directors, i.e. companies with directors appointed by controlling shareholders embrace better CSR strategies and provide better CSR reporting than companies with directors appointed by funds and funds representatives [73] and this can be observed in the case of Škoda and Čepro. In addition, Spanish companies were used to demonstrate that internal and external CSR contributes to the enhancement of the product and service quality and should be perceived and reported as such [74] and this can be observed as well in the case of Škoda. Regarding French companies and their CSR reporting, it has been observed a strong link between the CSR and branding, in particular that the store brand with price-quality ration contributes to the development of a positive price image and CSR image [75], exactly as in the case of ČEZ or Škoda. Indeed, outside of the EU, especially in the USA, South Korea and Japan, a very interesting relation is observed between the CSR drive and commitment, the use of luxury brands and powers of individuals [76] and undoubtedly Škoda and Hyundai would like to adhere to it. Nevertheless, strong voices presenting the CSR commitment as the win-win situation for shareholders and stakeholders [77] from the other side of the Atlantic are not (yet) echoed by the Czech reality and annual reports of (at least the selected) Czech Companies.

..

For sure, each annual report has the potential to convey a declaration to society, but this potential is not always developed. In addition, these declarations have in common only the adherence to international standardization. Otherwise, each company seems to exercise quite a bit of discretion about how they used the annual report and CSR information for information and marketing purposes. Further, it can be proposed that these trends described on the observed Czech companies reflect to certain extent patterns observed abroad, especially with respect to the Central Europe.

Point 4:

When the author relocates some of the paragraphs from the results section to other sections, he/she forgot to (re)number the references. For example, references [42, 57] in line 114 or references [58-62] in line 116 must not appear before references [37], [38], etc. Please, check them and read the manuscript guidelines for authors.

Response:

Yes, this was an oversight from me, I apologize for that and am grateful for your attention. I have carefully checked all references and made changes.

Point 5: Results

In addition, I still think that some of the information from the results section (lines 364-384) should be relocated as well, or it would be simply removed, because it is not directed linked with the results. It these paragraphs the author sums up again the information previously stated in other sections of the manuscript and, in my opinion, it is not necessary to duplicated it at all.

Response:

Well, I think this information has its place and relevance here. However, I see the point of the risk of repetitiveness and so as a compromise solution, I have reduced this part.

The new wording is as follows:

Nevertheless, one can argue that it implies a general rule that shareholder companies and limited liability companies with more than 500 employees have to include CSR information in their annual reports and file them with their national Commercial Registers.

The Czech law extends this general legal duty generated by the Directive 2013 and the Directive 2017 and demands electronic filing. This leads to the free availability of annual reports and ultimately becomes a part of the data available via BRIS. The selected 10 largest Czech companies satisfy the given criteria and are subject to this legal duty, i.e. their annual statements with the CSR information are to be available via BRIS.

It is extremely interesting to make a pioneering case study involving annual reports of the 10 Czech largest companies for 2013-2017 and conduct a research about whether they provide CSR information and, if yes, what kind (R&D, environmental protection, employment matters, others) and in what quantity and quality, and whether the very wording or its general spirit provides hints and indices able to be considered as declarations. The set of tables below addresses these questions and provides an insight that is truly original.

Reviewer 2 Report

In the first review round, I suggested to highlight the strategic role of companies’ disclosure and asked for a table that synthetize the different information required classified according the different regulatory source.

With regard to the first point, amendments just put a sentence in which this role is highlighted.

A table is not added to the paper, but a set of sentences that describe the differenti information required are added.

However, a table would have given a better  understanding of the issue.

The main weak point remains the methodology section. The paper still misses to take into account four dimensions of CSR (R&D, ENVIRONMENT, EMPLOYEES AND OTHERS) .

Author Response

Sustainability 394041

Corporate Social Responsibility Information in Annual Reports in the EU – A Czech Case Study

Responses to Reviewer 1, 2 and 3 - Comments – All previous changes highlighted in yellow, all new changes hightlighted in green.

Reviewer 2

Point 1:

I asked for a table that synthetize the different information required classified according the different regulatory source….However, a table would have given a better  understanding of the issue.

Response:

Well, I must admit that firstly I was not enthused about the proposition and did not prepare the new table. However, I have re-considered my prior reluctance and realized that this is an excellent proposition and significantly changed this part of my paper and added the required new table. I am very grateful for this peer review recommendation and the persistence of its author. Yes, the new table matches perfectly and makes the entire paper clearer and more academically robust. Perhaps my legal background makes me think that “everybody knows” but this is wrong because the interaction of the EU and national law is often complex and to have a clear table summarizing the exact source and content of legal duties is indispensable.

As stated above, the previously changed parts of my paper are highlighted yellow, the new changes in green – so you can see how your proposition has dramatically changed (and I think as well improved) my paper. For your convenience, I am enclosing the new table.

Table 1. Key provisions constituting the legal framework for CSR e-reporting in the Czech Republic.

Source

Content

Directive 2013

Art.19a(1) Non-financial statement

“1. Large undertakings which are public-interest entities exceeding on   their balance sheet dates the criterion of the average number of 500   employees during the financial year shall include in the management report a   non-financial statement containing information to the extent necessary for an   understanding of the undertaking's development, performance, position and   impact of its activity, relating to, as a minimum, environmental, social and   employee matters, respect for human rights, anti-corruption and bribery   matters,…” 

Directive 2017

Art.13 Scope 

“The coordination measures prescribed by this Section shall apply to the   laws, regulations and administrative provisions of the Member States relating   to the types of company listed in Annex II.”

Art.14 Documents and particulars to be disclosed by companies 

“Member States shall take the measures required to ensure compulsory   disclosure by companies of at least the following documents and   particulars…the accounting documents for each financial year which are   required to be published in accordance with Council Directives …”

Art.18 Availability of electronic copies of documents and   particulars 

“Electronic copies of the documents and particulars referred to in   Article 14 shall also be made publicly available through the system of   interconnection of registers”

Annex II Types of Companies referred to in Articles 7(1), 13, 29 …

— Czech Republic: společnost s ručením omezeným, akciová společnos   (private limited company – limited liability company (s.r.o./Ltd.) and public   limited company – shareholder company (a.s./SA).

Act 1991

Art.20 duty to have final statements verified by an auditor extends to   companies with at least 50 employees or turnover over 80 million CZK or …..

Art.21 companies with duty to have financial statements verified have as   well duty to prepare annual reports … Annual reports have to include both   financial and non financial information including about R&D, environment   and employement matters…

Act 2013

Art.2-3 Commercial Register along with Collection of documents are kept   by courts and are freely available in a digital format

Art.42 all companies and corporations to be registered with Commercial   Register

Art.66 document to be filed in the Collection of documents include annual   reports and final statements as stated in Act 1991

Point 2:

The main weak point remains the methodology section. The paper still misses to take into account four dimensions of CSR (R&D, ENVIRONMENT, EMPLOYEES AND OTHERS) .

Response:

Yes, I have significantly changed the methodology section and, among other changes, added information about these four dimensions (their source and reasons for their selection). There are many changes in the methodology section and with respect to the four dimensions, the most important change is this one:

. The data on CSR was classified by their type and these types were done based on Directive 2013, i.e. R&D, environmental protection, environmental matters and others. These four types are further described in the recital 26, Art.19a, Art.29a of the Directive 2013, in explanations provided by the European Commission [46] and in ISO 26000. Considering the focus on the Czech Republic, it is pivotal to underline that the Czech national law deals exactly with these four dimensions, see Act 1991 and especially its Art.21. Unfortunately, this Czech national provisions does not provide any further descriptions and merely satisfies itself by demanding annual reports with non financial statements about “activities in the field of R&D, activities in the field of environment protection and in employment relationships”. Hence these four dimensions are officially and explicitly recognized by the Czech law and the CSR information in annual reports has to entail them, but there is no clear rule about the expected exact extent and depth of this information and so it is used the reliance on the indication provided by the EU law (Directive 2013) with the above mentioned explanations and ISO norms .

Reviewer 3 Report

Dear Author,

I am very sorry but I am still confirming the major revision requested in the previous round of review. As you stated at the end of the review report, I think the revision that me and the other referees has suggested had not been fully addressed in the revised version of the paper.

I also would like to point out that all of us (the three referees) has the same concerns.

Therefore, I kindly invite you to strenghten the paper and take the suggested revision seriously

Author Response

Sustainability 394041

Corporate Social Responsibility Information in Annual Reports in the EU – A Czech Case Study

Responses to Reviewer 1, 2 and 3 - Comments – All previous changes highlighted in yellow, all new changes hightlighted in green.

Reviewer 3

I am very sorry but I am still confirming the major revision requested in the previous round of review. As you stated at the end of the review report, I think the revision that me and the other referees has suggested had not been fully addressed in the revised version of the paper.

I also would like to point out that all of us (the three referees) has the same concerns.

Therefore, I kindly invite you to strenghten the paper and take the suggested revision seriously

Response:

Yes, you are right and I have moved to dramatically change my paper – the methodology has been redone (Delphi, four dimensions, dual purpose explanation, etc.), a new table has been prepared and included, the inter-relation between case study and other findings has been strengthened, newly published papers were considered and referred. Consequently, the references have been re-numbered and the new version of my paper has undergone a new proofreading by a native speaker. I trust the new version will meet your basic expectation.

At the same time, yes, I admit that I have not been able to follow your very interesting proposition about the comparison with case studies from other EU countries. Being a lawyer, I feel strongly that firstly we would have to deal with a comparison of the legal frameworks, i.e. to identify whether the CSR reporting is not in the particular jurisdiction impacted by national local particularities. Only once this would be determined, could we compare and assess the differences between annual reports established and filed in various EU member states jurisdictions. After all, Directives 2013 and 2017 are “merely” Directives and not Regulations and, in addition, they do not seem to go for the full harmonization and so EU member states still have manoeuvring space about the exact setting of their national laws with respect to the CSR and annual reports. I must humbly admit that I do not master these national laws and so, at this point, could not correctly address your proposition. Nevertheless, I want to emphasize that this is a great idea and a huge challenge and perhaps in the future I will try to get in touch with the required experts from other jurisdictions and in collaboration with them attempt to prepare such a paper (these future co-authors could bring the knowledge about their national law and the data about their companies and their annual reports).

Hence, at this point I would like to ask for your benevolence to consider my current paper with its focus on only EU law and Czech law and only on 50 annual reports of 10 largest Czech companies.

Previous changes are highlighted in yellow, new changes are highlighted in green.

Thank you.

Round 3

Reviewer 1 Report

Dear author,

After revising the new version of the manuscript I have checked that all the suggestions made by the reviewers have been taken into account and the necessary changes to improve the paper have been made. 

So I my opinion the manuscript have the enough quality to be publish. 

Reviewer 3 Report

I appreaciated the improvements in the methodological section, even if no mention (and references) to content analysis has been provided. From my understanding, the analysis performed has been a content analysis and my first referee report provided complete references for this.